# Chemical and Quality Analysis of Beauty Tea Processed from Fresh Leaves of Tieguanyin Variety with Different Puncturing Degrees

**DOI:** 10.3390/foods12091737

**Published:** 2023-04-22

**Authors:** Mingjin Li, Yunzhi Zhang, Chunmei Chen, Sitong Zhong, Minxuan Li, Kai Xu, Yanyu Zhu, Pengchun Li, Shijun You, Shan Jin

**Affiliations:** 1College of Horticulture, Fujian Agriculture and Forestry University, Fuzhou 350002, China; limingjin0511@163.com (M.L.); zyz3133@163.com (Y.Z.); zhongsitong1110@163.com (S.Z.); xukai97@foxmail.com (K.X.); zyy8585065@163.com (Y.Z.); 2Fujian Fengyuan Tea Industry Co., Ltd., Sanming 366100, China; chenchunmei8887@126.com; 3College of Plant Protection, Fujian Agriculture and Forestry University, Fuzhou 350002, China; minxuan0512@163.com; 4Fujian Jiangshan Beauty Tea Co., Ltd., Sanming 366100, China; jsmrtea@163.com

**Keywords:** beauty tea, leafhopper puncturing degree, UPLC-Q-TOF/MS, GC-MS, metabolite, quality

## Abstract

Beauty tea with special flavor can be affected by the degree of leafhopper puncturing. The present research adopted widely targeted metabolomics to analyze the characteristic metabolites of fresh tea leaves and beauty tea with different degrees of leafhopper puncturing. Low-puncturing beauty tea (LPBT) exhibited a superior quality. Altogether, 95 and 65 differential metabolites, including tea polyphenols, saccharides, and lipids, were identified from fresh leaves and beauty tea, respectively. The partial least squares regression (PLSR) analysis results showed that isomaltulose, theaflavic acid, and ellagic acid, may be the characteristic metabolites that form the different taste outlines of beauty tea. Based on odor activity values (OAVs) and partial least squares discriminant analysis (PLS-DA), dihydrolinalool and cis-linalool oxide were identified as characteristic volatile components, which may be essential for the formation of the different aroma characteristic of beauty tea. The results provide a theoretical basis for selecting raw materials, performing quality research, and developing beauty tea industrially.

## 1. Introduction

Tieguanyin is the main oolong tea variety in Fujian Province and even in China. It is the first batch of excellent tea variety recognized in China and has become an important parent of oolong tea breeding. To date, the backbone parents of tea plant cross-breeding, Tieguanyin and Huangdan, have spawned 12 national or provincial varieties such as Jinguanyin (Minaceae No. 1), Huangguanyin, and Jinmudan [1], accounting for 52.6% of the total number of new varieties of oolong tea at the national and provincial levels [2].

Beauty tea, a deeply fermented variety of oolong tea (60–80% fermentation), which selects the fresh leaves of one-bud and two-leaf tea infected by tea green leafhopper as raw materials and is processed according to specific processing technology. The largest beauty tea production base in China is located in Datian County, Fujian Province, with a planting area of 70,000 mu and an output of 4000 tons in 2021, accounting for 70% of the national beauty tea output [3]. Tieguanyin (Figure 1), as the main variety in Datian County, can be processed into black tea that tastes mellow and smells sweet, and the processed Minnan oolong tea has the advantages of long-lasting fragrance, mellow and sweet taste, and enjoys the reputation of “Guan Yin Yun” [4]. However, the quality of Tieguanyin in making beauty tea is not ideal, and the products exhibit an uneven quality due to the different degrees of damage to fresh tea leaves by leafhopper puncturing. Therefore, it is an important way to improve the ratio of Tieguanyin to produce high-quality beauty tea to explore the raw materials of Tieguanyin fresh leaves which are conducive to the improvement of beauty tea quality by utilizing the different degree of leafhopper puncturing on fresh tea leaves.

The fresh tea leaves are punctured by tea green leafhoppers, which not only causes redness and scorch in the appearance of fresh leaves, but also causes changes in the composition of fresh leaves [5,6]. Different degrees of leafhopper puncturing produce different effects on the metabolic composition of tea, and a moderate degree of puncturing can cause tea to release a special flavor [7], but this degree may vary with varieties [8]. In this study, the fresh leaves of Tieguanyin tea variety with different puncturing degrees of tea green leafhoppers were used as raw materials to prepare beauty tea according to specific processing technology. Using the non-targeted metabolomics, which were based on ultra-high performance liquid chromatography-quadrupole time-of-flight mass spectrometry (UPLC-Q-TOF/MS) and gas chromatography-mass spectrometry (GC-MS), combined with sensory evaluation, the influence of the puncturing degree of tea green leafhopper on the metabolites of fresh tea leaves and beauty tea was studied. Furthermore, odor activity value (OAV) and multivariate statistical analysis were performed to reveal the different metabolites of the beauty tea with different degrees of puncturing. The results of this study will provide theoretical and practical guidance for the preparation of high-quality beauty tea ingredients and provide in-depth insight for determining the characteristic metabolites that contributed to forming the distinctive flavors of beauty tea with different degrees of leafhopper puncturing.

## 2. Materials and Methods

### 2.1. Chemicals

The caffeine (≥99%), epigallocatechin gallate (EGCG, ≥99%), epigallocatechin (EGC, ≥98%), L-theanine (≥98%), epicatechin (EC, ≥99%), gallocatechin (GC, ≥98%), catechin (C, ≥99%), epicatechin gallate (ECG, ≥98%) were purchased from Chengdu Munster Biotechnology Co., LTD. (Chengdu, China). The formic acid (≥99.0%), methanol (≥99.9%) and acetonitrile (≥99.9%) were purchased from Shanghai Yi En Chemical Technology Co., Ltd. (Shanghai, China) and were chromatographic grade.

### 2.2. Tea Samples

Fresh leaves with different degrees of puncturing were picked from Tieguanyin cultivar in the ecological tea garden (117°88′ E, 25°54′ N, ASL812.63 m) of Jingtian Tea Industry Co., Ltd., Datian County, Fujian Province, in August 2021. Part of the fresh leaves were immediately frozen and stored in liquid nitrogen at −80 °C for UPLC-Q-TOF/MS analysis after picking. The other parts were made into beauty tea according to the beauty tea processing technology (withering, tossing, fermentation, enzyme-inactivation, softening, rolling, drying), which were packed in vacuum-sealed bags for subsequent analysis.

The appearance of fresh leaves and beauty tea with different puncturing degrees is shown in Figure 2. Non-puncturing fresh tea leaves (NPFL) is when tea green leafhoppers do not cause damage and the growth of buds and leaves was good. Low-puncturing fresh tea leaves (LPFL) is when the leaf becomes yellow, stiff, and curled; the edge is slightly curled, or there are obvious outward or inward folds; and the veins become red. High-puncturing fresh tea leaves (HPFL) is when severe insect damage, severe leaf curl, burnt edges, or severe burnt edges are observed. The three kinds of fresh leaves were processed into non-puncturing beauty tea (NPBT), low-puncturing beauty tea (LPBT), and high-puncturing beauty tea (HPBT), respectively.

### 2.3. Sensory Evaluation

Beauty teas were evaluated according to GB/T23776-2018. A quantity of 5 g of each tea sample was weighed and placed in a 110 mL evaluation bowl. The tea was brewed with boiling water for 2 min, 3 min, and 5 min respectively. Five sensory evaluation experts (three female and two male, 35–55 years old) from the Fujian Agriculture and Forestry University gave descriptions and scores, according to the description standard of “Beauty Tea Brewing and Tasting” (T-CSTEA00006-2019). Then, panelists conducted sensory tests on the intensity values (0–10) of taste attributes [9] (mellow, thick, astringency, brisk, bitter, sweet, umami) of each beauty tea infusion. All reviewers are professionally trained and have more than five years of experience in sensory analysis.

### 2.4. UPLC-Q-TOF/MS Analysis

The non-volatile metabolites were determined by UPLC-Q-TOF/MS (Waters, Manchester, UK). The freeze-dried tea samples were ground into powder with a TissueLyser II (QIAGEN, Hilden, Germany), accurately weighed to 60 mg and placed in a 2 mL EP tube, then extracted with 2 mL 60% methanol. After ultrasonic extraction (35 °C, 40 min), centrifugation (12,000 rpm, 4 °C) for 10 min, supernatant was collected, diluted five times, and then a 0.22 μm filter membrane was used for UPLC-Q-TOF/MS detection. The experiment was repeated using four biological methods.

The acquity UPLC shield BEHC_18_ column (1.7 μm, 2.1 mm × 100 mm, Waters, Wexford, Ireland) was used for the non-volatile metabolites. The mobile phase consisted of 0.1% formic acid in water (*V*/*V*) (A) and 0.1% formic acid in acetonitrile (*V*/*V*) (B). The elution gradient was as follows: 0 min: 100% A; 2 min: 93% A; 13 min: 60% A; 18–19 min: 100% B; 21–24 min: 100% A. The injection volume was 1.0 μL. The temperatures of sample manager and column were set at 4 °C and 40 °C, respectively.

The parameters of time-of-flight mass spectrometry were set as follows: detection mode ESI^−^ and ESI^+^; capillary voltage 2.80 kV; high energy ramp 20–30 eV; scanning range 50–1000 *m*/*z*; source temperature 120 °C; desolvation temperature 380 °C. The metabolites were identified by comparing them with KEGG, Massbank, HMDB, and other databases, standards, and related literatures.

### 2.5. GC-MS Analysis

A 7980B-5977A gas chromatographic-mass spectrometry (Agilent, Santa Clara, CA, USA) equipped with Agilent HP-5MS column (30 m × 250 μm × 0.25 μm, Agilent, Santa Clara, CA, USA) was used for the volatile compounds analyses. Refer to the experimental method of our research group [10] for the aroma-extraction and GC-MS analytical procedure. The volatile peaks were determined by matching the mass spectrometry database of the National Institute of Standards and Technology 11 (NIST11) (match degree ≥ 70) and referring to relevant literature on tea aroma compounds. The chemical structures and names of the volatile compounds according to the NIST chemical network (https://webbook.nist.gov/chemistry/cas-ser/, accessed on 26 April 2022) and PubChem (https://pubchem.ncbi.nlm.nih.gov, accessed on 26 April 2022). The odor description of aroma was obtained by referring to relevant literature. The relative content (μg/kg) of volatile constituents was calculated on the internal standard method [10]. Three biological replicates were performed on the test samples.

### 2.6. Odor Activity Values (OAVs) Calculation

Refer to Xu et al. [9]. The calculation formula for *OAV* value is:*OAV_i_ = C_i_/OT_i_*(1)

Note: *C_i_* (μg/kg) was the content of the volatile compounds; *OT_i_* (μg/kg) was the aroma threshold of volatile components in water.

### 2.7. Statistical Analysis

All data were presented as mean ± standard statistical (SD) of three replicates and calculated by Microsoft Excel 2016. The significance analysis was carried out with one-way ANOVA and Tukey’s multiple range test using SPSS 20.0 (Armonk, NY, USA). The principal component analysis (PCA), partial least squares discrimination analysis (PLS-DA) and partial least squares regression (PLSR) were performed online (https://www.metaboanalyst.ca, accessed on 2 December 2022) and using SIMCA-P 13.0 software (Umetrics, Umea, Sweden). The cluster heatmap and Venn plots were generated using TBtools 1.0971 (South China Agricultural University, Guangzhou, China).

## 3. Result

### 3.1. Sensory Evaluation

To compare the quality of the three beauty teas (NPBT, LPBT, and HPBT), the appearance and endoplasms of beauty tea were evaluated and scored by sensory recognition panelists. The statistical significance among the different qualities of beauty tea was determined by one-way ANOVA and Tukey’s multiple range test (*p* < 0.05). The sensory evaluation (Table 1) showed that LPBT had the best quality, the quality of NPBT was the worst. This indicated that the quality of beauty tea processed from low-puncturing fresh tea leaves was better than that high-puncturing. The appearance was red-auburn in all the beauty tea, with gradings of little dull, little curly, approach tippy, and even in NPBT, yellow and white, curly, tippy, and even in LPBT, and yellow and white, more curly, more tippy, and more even in HPBT. A bright liquor color was defined in all the beauty tea and presented as orange-red, deeply orange, and amber with increasing puncturing degree. The infused leaf was uniform in all the beauty tea, and the HPBT was the most curly, but the LPBT was softer (Figure 3). Furthermore, there were significant differences in aroma and taste between NPBT, LPBT, and HPBT. In terms of aroma, compared with the obvious floral aroma of NPBT, the aromas of LPBT and HPBT were richer, both with honey fragrance, but LPBT was more fragrant and lasting. In terms of taste, NPBT had a strong, thick, and slightly bitter taste, while LPBT and HPBT had a sweet, brisk, mellow and thick taste. The results indicated that LP degree exhibited an promoting effect on the quality of beauty tea. Therefore, the quality of beauty tea is superior when the fresh leaves are lightly punctured by tea green leafhoppers.

### 3.2. Analysis of the Non-Volatile Differential Metabolites in Fresh Tea Leaves

#### 3.2.1. Non-Volatile Metabolite Profiling of Fresh Leaves

To explore the differences of non-volatile metabolites in NPFL, LPFL, and HPFL, multivariate statistical analysis was performed. A PCA score plot (Figure 4A) showed a clear separation of the NPFL, LPFL, and HPFL samples in total ion mode, the PC1 explained 70.7% of the variation, PC2 explained 15.4% of the variation, and PC3 explained 10.2% of the variation, indicating that there were differences in the non-volatile metabolites of NPFL, LPFL, and HPFL. The supervised analysis method PLS-DA was used to investigate the differences between the NPFL, LPFL, and HPFL. The PLS-DA score plot (Figure 4B) showed that they had differences, similar to the results of PCA. The permutation plots (Figure 4C) of PLS-DA showed that all blue Q2-values to the left were lower than the original points to the right and the blue regression line of the Q2-points intersects the vertical axis (on the left) below zero, indicating the model had an effective predictive ability, without overfitting.

#### 3.2.2. Identification and Analysis of Differential Metabolites in Fresh Leaves

The *p* < 0.05 and variable importance in the project (VIP) > 1 were the screening conditions for differential metabolites. A total of 95 differential metabolites were identified, including 13 categories, as shown in Appendix A. These were 12 catechins and their derivatives, 40 flavones and flavonols and their glycosides, 8 phenolic acids, 1 amino acid, 2 organic acids, 6 proanthocyanidins, 1 theaflavin, 2 coumarins, 9 saccharides and glycoside derivatives, 3 linalool glycosides, 4 tannins, 6 lipids, and 1 other metabolite, indicating that the degree of leafhopper puncturing mainly affected flavones and flavonols and their glycosides, catechins and their derivatives, saccharides and glycoside derivatives, phenolic acids, proanthocyanidins, and lipids. According to the relative content of metabolites, the cluster heatmap analysis was conducted on them (Figure 5). The heatmap showed large differences in metabolite abundances between NPFL and HPFL, but small differences between LPFL and NPFL, which were consistent with the result of PCA. Compared with NPFL, catechins and their derivatives, flavones and flavonols and their glycosides, phenolic acids, organic acids, theaflavin, saccharides, and glycoside derivatives showed decreased levels in LPFL, while amino acid, coumarins, and lipids showed increased levels. Catechins and their derivatives, amino acid, proanthocyanidins, theaflavin, coumarins, linalool glycosides, and tannins in HPFL showed increased levels, while flavones and flavonols and their glycosides, phenolic acids, organic acids, saccharides and glycoside derivatives, and lipids showed decreased levels. In addition, the contents of catechins and their derivatives, flavones and flavonols and their glycosides, amino acid, prothanocyanidins, theaflavin, linalool glycosides, and tannins in LPFL were lower than HPFL, and the contents of phenolic acids, organic acids, saccharides and glycoside derivatives and lipids were higher than HPFL. The contents of most metabolites in fresh leaves decreased after being punctured by tea green leafhoppers. These differential metabolites may be important components in the identification of fresh tea leaves with different degrees of puncturing.

#### 3.2.3. Analysis of Biosynthetic Pathways of Differential Metabolites

The differential metabolites in the fresh tea leaves were further analyzed by the KEGG pathway-enrichment method (KEGG, www.geno.me.jp/kegg/pathway.html#metabolism, accessed on 24 April 2022). The results showed that flavonoid biosynthesis (three metabolites) and flavone and flavonol biosynthesis (two metabolites) were the main enrichment pathways. Many of the products of the flavonoid biosynthesis and flavone and flavonol biosynthesis pathways were essential for tea quality, especially taste and color of the tea infusion. Compared with NPFL samples, the metabolites of (−)−EGC, (−)−GC, and naringin involved in flavonoid biosynthesis and the metabolites of nictoflorin and astragalin involved in flavone and flavonol biosynthesis were down-regulated in LPFL and HPFL samples (Figure 6). As these metabolites are essential for anthocyanin synthesis, the difference in anthocyanin content may be the cause of the color differences between NPFL, LPFL, and HPFL.

### 3.3. Analysis of Taste Components of Beauty Tea

#### 3.3.1. Metabolomic Analysis of Non-Volatile Metabolites

The multivariate statistical analysis was also carried out. A PCA score plot (Figure 7A) showed a clear separation of the NPBT, LPBT and HPBT samples in total ion mode, indicating that there were differences in non-volatile metabolites of NPBT, LPBT, and HPBT. The PLS-DA score plot (Figure 7B) showed that they had differences in the non-volatile components of three beauty teas, which was similar to the results of PCA. The permutation plot (Figure 7C) of PLS-DA showed that the model had effective predictive ability and did not experience overfitting.

#### 3.3.2. Identification and Analysis of Non-Volatile Metabolites

The *p* < 0.05 and VIP > 1 were the screening conditions for differential metabolites. A total of 65 differential metabolites including 12 classes were identified, as shown in Appendix A. These were 11 catechins and their derivatives, 17 flavones and flavonols and their glycoside, 4 phenolic acids, 1 amino acid, 2 organic acids, 4 prothanocyanidins, 5 theaflavins, 10 saccharides and glycoside derivatives, 4 linalool glycosides, 3 tannins, 3 lipids, and 1 other metabolite, indicating that tea polyphenols, saccharides and glycoside derivatives, and linalool glycosides may be the basis for forming the taste characteristics of beauty tea with different puncturing degrees. According to the cluster heatmap (Figure 8), it showed that the metabolite abundances of NPBT and HPBT differ greatly, while those of LPBT and HPBT differ little, which was consistent with the result of PCA. Compared with NPBT, flavones and flavonols and their glycosides, phenolic acids, proanthocyanidins, theaflavins, tannins and lipids in LPBT showed decreased levels, while amino acid, organic acids, saccharides and glycoside derivatives, and linalool glycosides showed increased level. Catechins and their derivatives, flavones and flavonols and their glycosides, phenolic acids, amino acid, proanthocyanidins, theaflavins, and tannins in HPBT showed decreased levels, while organic acids, saccharides and glycoside derivatives, linalool glycosides, and lipids showed increased levels. In addition, the contents of catechins and their derivatives, flavones and flavonols and their glycosides, phenolic acids, amino acid, proanthocyanidins, and tannins in LPBT were higher than HPBT, while organic acids, saccharides and glycoside derivatives, linalool glycosides, and lipids were lower than HPBT. In general, the total content of metabolites in beauty tea decreased as puncturing degree increased.

#### 3.3.3. Correlation Analysis between Taste Profiles and Non-Volatile Differential Metabolites of Beauty Tea

To understand the taste properties of beauty tea, a sensory test on the intensity values (0–10) of taste attributes (mellow, thick, astringency, brisk, bitter, sweet, umami) of each beauty tea infusion was conducted by sensory recognition panelists, and the taste attribute scores of the three beauty teas were compared by a sensory panel (Figure 9A). Higher mellowness and sweetness were found in HPBT, followed by LPBT, and finally by NPBT, whereas the thickness, astringency, and bitterness were the opposite. Additionally, LPBT had stronger briskness and umami than those in NPBT and HPBT, higher briskness was found in HPBT, but NPBT had stronger umami than that in HPBT.

To further understand and ascertain the characteristic non-volatile metabolites contributing to the taste attributes of the three beauty teas, the PLSR model with 65 non-volatile differential metabolites (X) and taste attribute scores (Y) was performed (Figure 9B). Isomaltulose (M47; coefficient: 0.051, 0.048), 12-O-Î^2^-D-glucopyranosyloxyjasmonic acid (M34; coefficient: 0.050, 0.049) and linalool oxide D 3-[apiosyl-(1→6)-glucoside] (M55; coefficient: 0.042, 0.044) were correlated to mellowness and sweetness, while achimilic acid (M35, coefficient: 0.040) and linalool 3,6-oxide primeveroside (M56, coefficient: 0.039) were also correlated to sweetness. Theaflavic acid (M59; coefficient: 0.042~0.046) and ellagic acid (M32, coefficient: 0.035~0.037) made a positive contribution to the bitterness, astringency, and thickness. Additionally, the dimerecatechins such as procyanidin A2 (M10, coefficient:0.060), EC-(4β→8)-EGC 3-O-gallate (M8, coefficient: 0.059), EC 3-O-gallate-(4β→6)-EGC 3-O-gallate(M7, coefficient: 0.056), EGC-(4β→8)-EC 3-O-gallate (M11, coefficient: 0.055), neotheaflavin 3-gallate (M43, coefficient: 0.057), and L-theanine (M33, coefficient: 0.045) had a positive contribution to the umami.

### 3.4. Analysis of Aroma Constituents of Beauty Tea

#### 3.4.1. Determination of Volatile Constituents in Beauty Tea

A total of 49 volatile components, including 6 alcohols, 4 aldehydes, 12 esters, 12 alkenes, 2 terpenes, 6 alkanes, 3 aromatics, 2 ketones, 1 furan, and 1 other, were authenticated by GC-MS in three beauty teas (Table 2). Among the 49 volatile components, 42 were detected in NPBT, 35 were detected in LPBT, and only 28 were detected in HPBT, indicating that the amount of volatiles in NPBT, LPBT, and HPBT might be one of the reasons for the difference in aroma quality of the three beauty teas. To further study the volatile components of the three beauty teas, the common and unique compounds of the three beauty teas were analyzed (Figure 10D). There were 23 common compounds: 9, 3, and 4 unique volatile compounds in NPBT, LPBT, and HPBT, respectively, indicating that the unique volatile components may be one of the reasons for the difference in aroma quality of the three beauty teas.

To compare the contents of various volatile components in the three beauty teas, we calculated the relative contents of 49 volatile components (Table 2), and the statistical significance among the different contents of volatile compounds in beauty tea was determined by one-way ANOVA and Tukey’s multiple range test (*p* < 0.05). The contents of dihydrolinalool and benzaldehyde were the highest in all beauty teas. Apart from dihydrolinalool and benzaldehyde, the contents of linalool and 2-methyl-butanoic acid hexyl ester were the highest in NPBT and LPBT. The ethyl 2-(5-methyl-5-vinyltetrahydrofuran-2-yl)propan-2-yl and linalool were the highest in HPBT. These results indicated that the difference of volatile components content might lead to the different aroma characteristics of beauty tea with different degrees of puncturing.

#### 3.4.2. Characteristic Analysis of Volatile Components

To distinguish characteristic of volatile components in the three beauty teas, multivariate data analyses were performed to reveal the differences in volatiles. A PCA score plot (Figure 10) showed a clear separation of the NPBT, LPBT and HPBT samples, indicating that there were differences in the volatile components of NPBT, LPBT and HPBT. The PLS-DA score plot (Figure 10B) showed that there were differences in the volatile components of beauty tea with different puncturing degrees, which were similar to the results of PCA. Furthermore, we found that R2Y and Q2 were 0.994 and 0.976, respectively, and the intercepts of Q2 and *Y* axis were less than 0, which could be interpreted as the effective predictive ability of the model and without overfitting (Figure 10C).

#### 3.4.3. Analysis of the Characteristic Volatile Components

The contribution of aroma compounds is influenced not only by their content, but also by their threshold. It is generally believed that aroma compounds with OAV ≥ 1 contribute significantly to the total aroma. According to the quantitative data and threshold values in Table 2, the OAVs could be calculated, and a total of 10 volatile components with OAV ≥ 1 were determined (Appendix A). These were linalool, geraniol, β-cyclocitral, benzeneacetaldehyde, methyl salicylate, dihydrolinalool, cis-linalol oxide, decanal, 2-pentyl-furan, and cedrol. Subsequently, the VIP values of volatile components (Figure 10E) were calculated. The volatiles with both VIP > 1 and OAVs ≥1 were selected as characteristic volatile compounds, and relevant published data in the literature and databases were used for further identification. Dihydrolinalool and cis-linalool oxide were identified as characteristic compounds.

## 4. Discussion

### 4.1. Analysis of Non-Volatile Components in Fresh Leaves with Different Puncturing Degrees

The fresh tea leaves are punctured by tea green leafhoppers, which not only causes redness and scorch in the appearance of fresh leaves, but also causes changes in the composition of fresh leaves. In this study, we analyzed the non-volatile compounds in the NPFL, LPFL and HPFL by the UPLC-Q-TOF/MS and found that there were differences in the metabolites of NPFL, LPFL, and HPFL (Figure 4A,B), indicating that the puncturing degree of the tea green leafhoppers affected the non-volatile metabolites in the fresh tea leaves.

In this study, the degree of leafhopper puncturing mainly affected the synthesis of tea polyphenols, saccharides and glycoside derivatives, and lipids. Catechin is the main compound of tea polyphenols, and biological stress can lead to the decrease of catechin content in fresh tea leaves [14]. In the process of infection, catechin is hydrolyzed, and gallic acid generated by hydrolysis can prevent infection of tea plants [15]. In this study, the contents of monocatechin and its derivatives in fresh tea leaves decreased with the increase of puncturing degree, while the dimeric catechin’s behavior was the opposite, indicating that monocatechin and its derivatives may be hydrolyzed to enhance insect resistance, and polymerization into dimeric catechin during the hydrolysis process. By means of metabonomics, a previous study found [16] that the content of flavonoids increased after infection of tea green leafhoppers. However, in this study, the content of flavones or flavonols and their glycosides in LPBT and HPBT was lower than that in NPBT, which may be due to the interruption of glucose metabolism and lack of glycoylation or impaired chloroplast function caused by leafhoppers [17]. Proanthocyanidins can also protect plants from pathogens and herbivores [18,19,20]. In this study, the content of proanthocyanidins increased in HPFL, indicating that proanthocyanidins had insect resistance. Saccharides are an important ingredient in providing energy. Studies found [21,22] that saccharides accumulate under drought stress to ensure adequate carbohydrate supply. In this study, the content of saccharides and glycoside derivatives decreased with the increase of the puncturing degree, suggesting that these components may be hydrolyzed for energy supply under biological stress. Another study found [23] that chlorogenic acid and other phenolic acids in peanuts increased in response to the infestation of lepidoptera pests. However, in this study, the content of phenolic acid decreased with the increase of the puncturing degree, which indicated that different crops may have different defense mechanisms stimulated by pests, and thus the metabolites change differently. In addition, different herbivores caused different changes in metabolites in tea plants [15].

### 4.2. Analysis on Taste Components of Beauty Tea with Different Puncturing Degrees

Taste is one of the primary factors in evaluating tea quality. Catechins, flavonoids, amino acids, soluble sugars, and alkaloids in tea directly or indirectly affect the taste of tea liquor [24]. It is worth noting that these metabolites affecting the taste and aroma of tea are significantly affected by the puncturing degree of tea green leafhoppers.

According to the result of sensory evaluation, the flavor of beauty tea with different puncturing degrees was different. The formation of different tea flavors is closely related to the type and content of metabolites [25]. To affirm the characteristics of non-volatile metabolites that affect the taste difference of beauty tea, a total of 65 different metabolites were identified in three beauty teas (VIP > 1, *p* < 0.05) (Appendix A). Among them, tea polyphenols, saccharides and glycoside derivatives, and linalool glycosides were the main differential metabolites. The levels of these metabolites in the three beauty teas had an obvious difference, indicating that the taste difference of NPBT, LPBT, and HPBT might be mainly caused by the difference of these non-volatile components. Subsequently, to further determine the specific contributions of non-volatile metabolites in the three beauty teas on taste attributes, the PLSR analysis was used to study the correlation between taste properties and 65 non-volatile compounds of beauty tea. The result of PLS regression analysis showed that isomaltulose and organic acids such as 12-O-Î^2^-D-glucopyranosyloxyjasmonic acid and achimilic acid had strong correlations with sweetness and mellowness. Isomaltulose, which belongs to the soluble sugars, contributed to the sweetness and mellowness of tea liquor [26,27]. Organic acids can inhibit the bitter taste of tea liquor [28] and may facilitate the formation of sweetness and mellowness. In this study, HPBT had higher organic acid levels, indicating that this may account for the mellower and sweeter taste of HPBT compared with NPBT and LPBT. Theaflavic acid and ellagic acid were associated with bitterness, astringency, and thickness. Theaflavic acid and ellagic acid, which belong to the tannins and phenolic acids, respectively, contributed to the bitterness and astringency [29,30] of tea liquor. In this study, the contents of theaflavic acid and ellagic acid in NPBT were significantly higher (*p* < 0.05) than that in LPBT and HPBT, indicating that this may be the reason why NPBT tasted more bitter than LPBT and HPBT. Additionally, L-theanine and many dimerecatechins such as neotheaflavin 3-gallate, procyanidin A2, and EC-(4β→8)-EGC 3-O-gallate contributed to the umami, which may be the pivotal compounds responsible for the higher umami of NPBT and LPBT compared with HPBT.

Interestingly, we also found that the precursor of aroma, including linalool oxide D 3-[apiosyl-(1→6)-glucoside] and linalool 3,6-oxide primeveroside, correlated to mellowness and sweetness. These metabolites will release aroma compounds during tea manufacturing and brewing [31]. In this study, HPBT had higher contents of linalool oxide D 3-[apiosyl-(1→6)-glucoside] and linalool 3,6-oxide primeveroside, suggesting that these metabolites also contribute to the formation of the sweet and mellow taste of HPBT.

### 4.3. Analysis on Aroma Components of Beauty Tea with Different Puncturing Degrees

There are many factors affecting the aroma quality of tea, such as tea variety, processing technology, growth conditions and exogenous induction, and the change of aroma quality caused by tea green leafhoppers infecting tea leaves belongs to exogenous induction. The types and contents of volatile components were closely related to the formation of aroma characteristics of beauty tea and were greatly affected by the degree of leafhopper puncturing. In addition to the type and content of volatile components, the characteristic aroma of tea also depends on aroma activity values and their synergistic effects. Therefore, dihydrolinalool and cis-linalool oxide with both VIP > 1 and OAVs ≥ 1 were identified as characteristic volatile components, indicating that dihydrolinalool and cis-linalool oxide may be essential for the formation of the unique aroma quality of beauty tea with different degrees of puncturing.

Dihydrolinalool is an important contribution to flowery and fruity aroma. It is a thermogenic compound that is also released by glycosides [32]. In this study, the content of dihydrolinalool in HPBT was significantly higher (*p* < 0.05) than in NPBT and LPBT. However, none of the three beauty teas were fruity and the floral aroma in HPBT was lower than NPBT and LPBT, suggesting that dihydrolinalool and some volatiles that did not resemble in odor or structure may have masking effect. Linalool can be metabolized into four linalool oxides, namely, cis-linalool oxide (furanoid) (linalool oxide I), trans- linalool oxide (furanoid) (linalool oxide II), cis- linalool oxide (pyranoid) (linalool oxide III), and trans- linalool oxide (pyranoid)(linalool oxide IV). A study found [33] that there was an obvious negative correlation between the content of linalool and linalool oxide I, but no significant negative correlations between the content of linalool and linalool oxide II. The difference of the invertase quantity of linalool oxide I and linalool oxide II may be the cause of this phenomenon. This also explained that there was no significant correlation between the contents of linalool and cis-linalool oxide in beauty tea with different puncturing degrees. However, in addition to being converted from linalool, linalool oxides can also be converted to glycoside binding forms. Because of their high stability, the volatile components of glycosides are abundant in tea plants [34]. A study found [35] that the sweet, flowery and fruity aroma of linalool oxide did not come from the oxidation of linalool, but from the glycoside form of linalool oxide in fresh tea leaves. In this study, four glycoside-binding linalool oxides, including linalool oxide d 3-[apiosyl-(1→6)-glucoside], linalool 3,6-oxide primeveroside, linalool 3,7-oxide β-primeveroside, and l-linalool 3-[xylosyl-(1→6)-glucoside], were determined in fresh tea leaves and dry beauty tea by UPLC-Q-TOF/MS. These glycoside-binding linalool oxides are considered aroma precursors and release aroma compounds during tea manufacturing and brewing [31]. It has been reported [33] that CsGHs and CsGTs are involved in the interconversion of glycoside-binding volatiles and free volatiles. In addition, CsGT may be related to the resistance of tea plants to environmental stresses such as insects and cold stress [36]. In this study, the content of cis-linalool oxide in HPBT was significantly higher (*p* < 0.05) than that in NPBT and LPBT, suggesting that cis-linalool oxide may be a pivotal contributor responsible for the flowery and honey aroma of HPBT. In addition, the change regularity of linalool glycoside content in fresh leaves and beauty tea and cis-linalool oxide content in beauty tea showed the same, indicating that the puncturing degree of leafhopper affected the content of linalool glycoside in fresh tea leaves, and then affected the level of cis-linalool oxide in beauty tea.

## 5. Conclusions

Our results showed that the metabolites in beauty tea were affected by different puncturing degrees of tea green leafhopper. A low puncturing degree of fresh tea leaves is more conducive to improving the quality of beauty tea. The degree of leafhopper puncturing affects the isomaltulose, 12-O-Î^2^-D-glucopyranosyloxyjasmonic acid, achimilic acid, theaflavic acid, ellagic acid, procyanidin A2, EC-(4β→8)-EGC 3-O-gallate, EC 3-O-gallate-(4β→6)-EGC 3-O-gallate, EGC-(4β→8)-EC 3-O-gallate, neotheaflavin 3-gallate, L-theanine, linalool 3,6-oxide primeveroside, and linalool oxide D 3-[apiosyl-(1→6)-glucoside] levels, thus forming the different taste outlines of beauty tea. In addition, the dihydrolinalool and cis-linalool oxide levels may be mainly affected by the degree of leafhopper puncturing, thus forming the different aroma profile of beauty tea. However, the major limitation of this study is that the number of degrees of leafhopper puncturing is small, and each puncturing degree is difficult to control. Therefore, there is a need to narrow the gap between the puncturing degrees of tea green leafhoppers, increase the number of puncturing degrees, and further determine the degree of puncturing which is most conducive to the quality of beauty tea.

## Figures and Tables

**Figure 1 foods-12-01737-f001:**
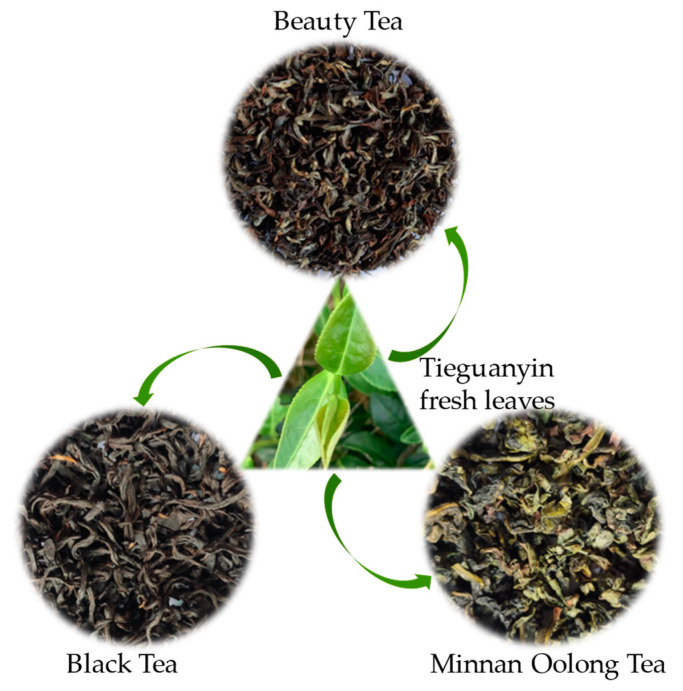
A variety of tea processed from Tieguanyin variety.

**Figure 2 foods-12-01737-f002:**
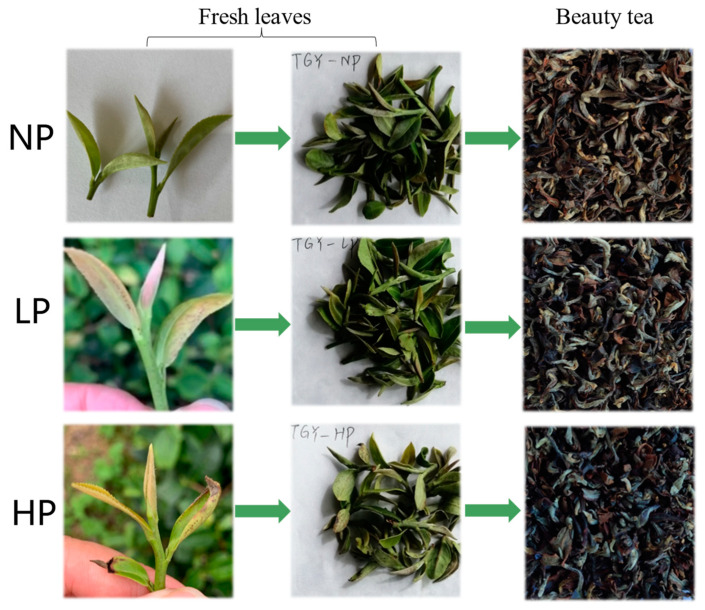
The appearance of fresh leaves and beauty tea with different puncturing degrees. NP means non-puncturing group; LP means low-puncturing group; HP means high-puncturing group.

**Figure 3 foods-12-01737-f003:**
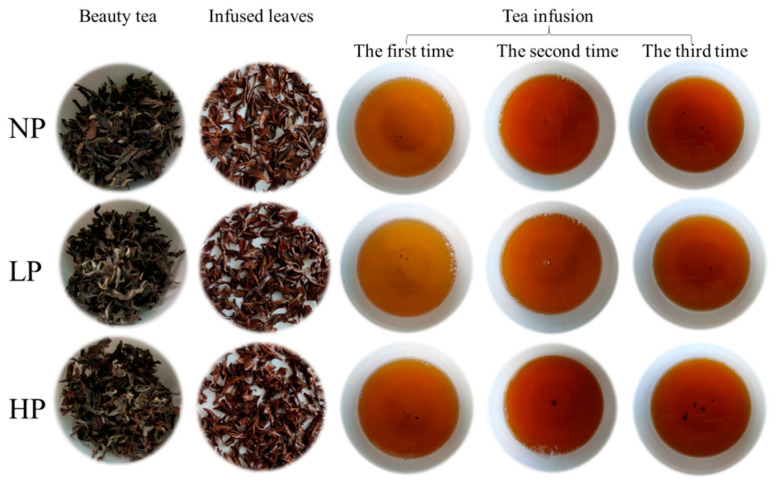
The appearance and infusion colors of beauty tea. NP means non-puncturing group; LP means low-puncturing group; HP means high-puncturing group.

**Figure 4 foods-12-01737-f004:**
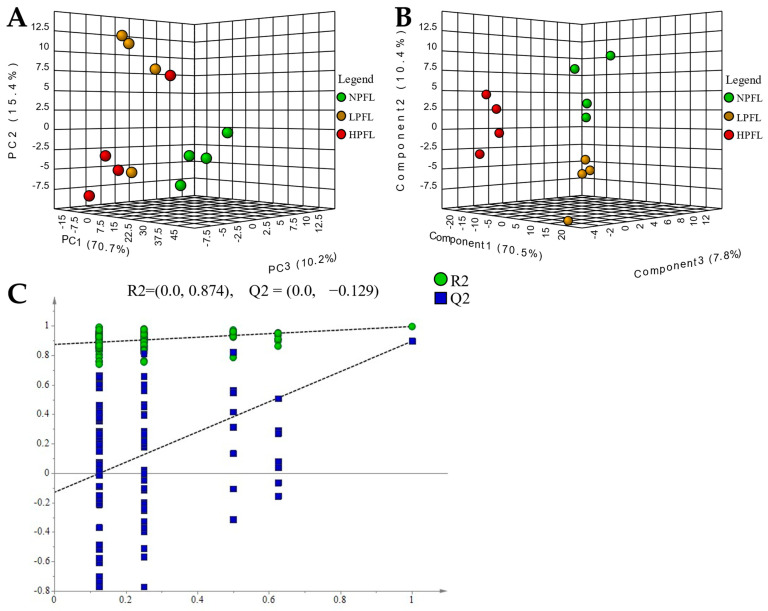
(**A**) principal component analysis (PCA)-X scores of fresh leaves; (**B**) partial least squares discriminant analysis (PLS-DA) score of fresh leaves; (**C**) PLS-DA model validation of fresh leaves.

**Figure 5 foods-12-01737-f005:**
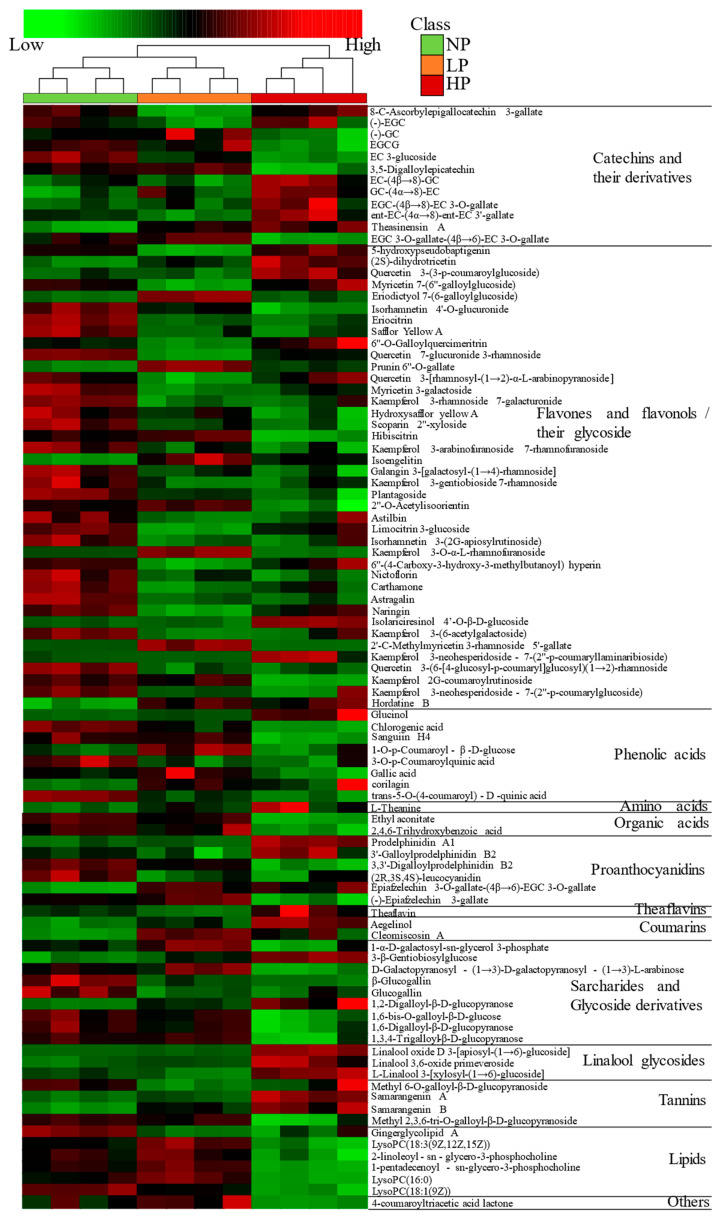
A hierarchical clustering heatmap of differential metabolites of fresh leaves.

**Figure 6 foods-12-01737-f006:**
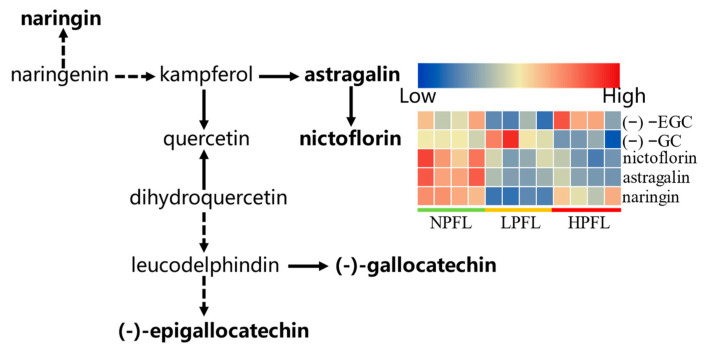
Differential metabolites involved in flavonoids metabolic pathway. “⇢” means that some metabolites were left out. Bold indicated the differential metabolites assigned to metabolic pathways.

**Figure 7 foods-12-01737-f007:**
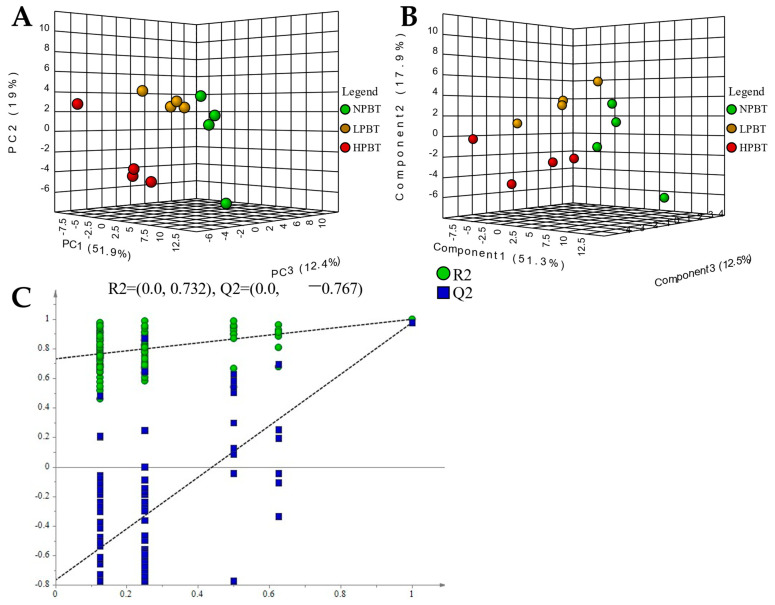
(**A**) PCA-X score of beauty tea; (**B**) PLS-DA score of beauty tea; (**C**) PLS-DA model validation of beauty tea.

**Figure 8 foods-12-01737-f008:**
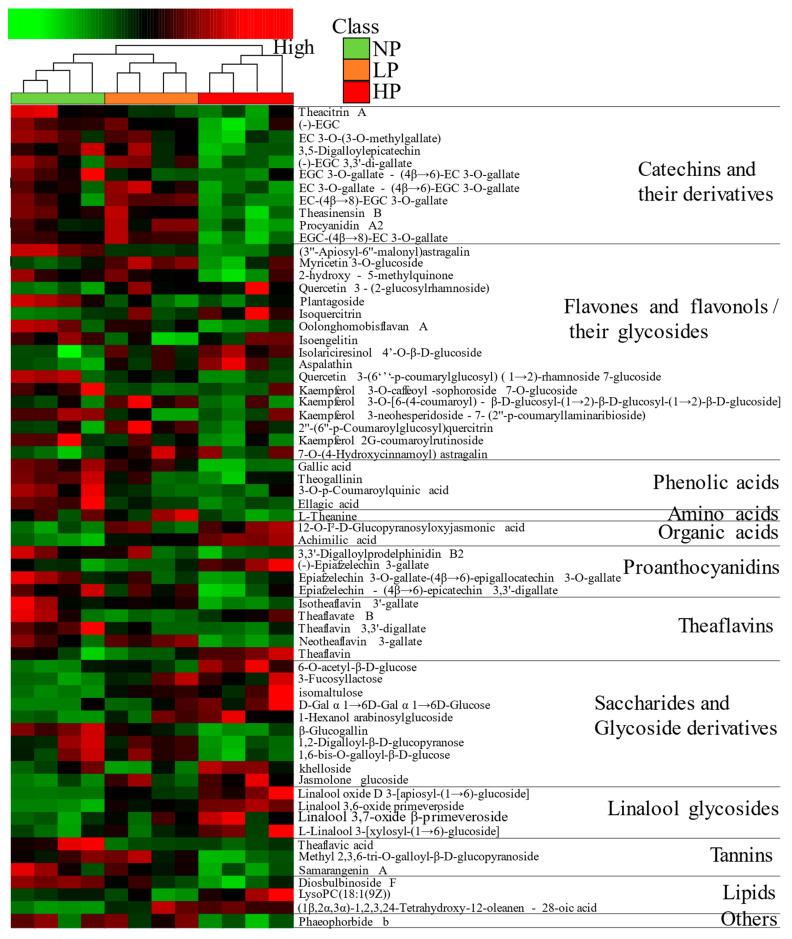
A hierarchical clustering heatmap of differential metabolites of beauty tea.

**Figure 9 foods-12-01737-f009:**
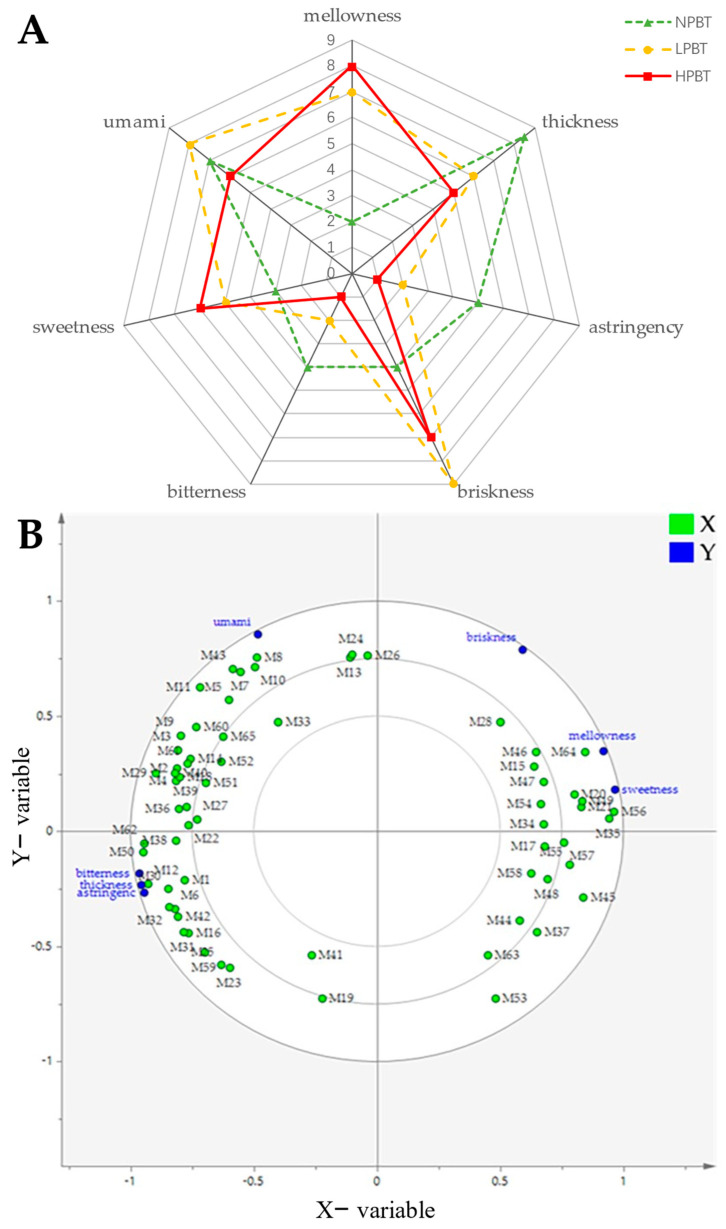
(**A**) Radar plots of taste profiles of beauty tea with different puncturing degrees; (**B**) partial least squares regression (PLSR) analysis between taste attributes and non-volatile differential metabolites (VIP > 1 and *p* < 0.05) of beauty tea with different puncturing degrees. *X*-axis: the 65 non-volatile differential metabolites (VIP > 1 and *p* < 0.05); *Y*-axis: the taste attribute scores.

**Figure 10 foods-12-01737-f010:**
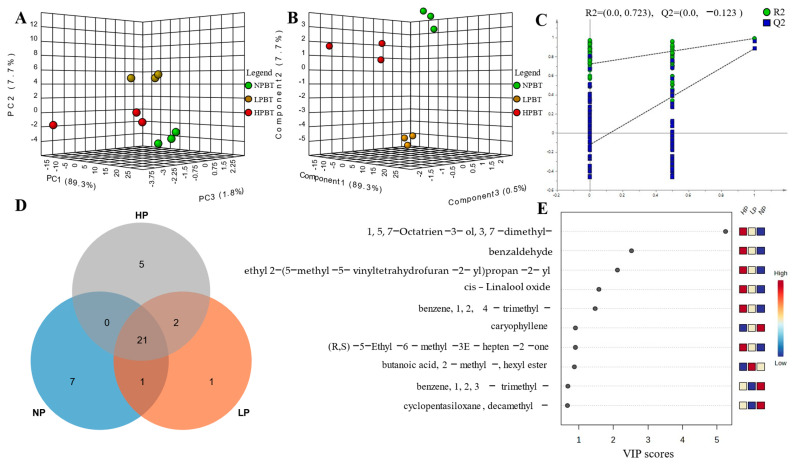
(**A**) PCA-X scores of beauty tea; (**B**) The PLS-DA scores for the 49 volatile components of beauty tea; (**C**) permutation test plots of the 49 volatile components in beauty tea; (**D**) Venn plot of beauty tea with different degrees of puncturing; (**E**) Variable importance in the project (VIP) plot of PLS-DA of beauty tea.

**Table 1 foods-12-01737-t001:** Sensory evaluation of beauty tea with different puncturing degrees.

Samples	Appearance (20%)		Aroma (30%)		Taste (35%)		Liquor Color (5%)		Infused Leaf (10%)		Total Score
	Comment	Score	Comment	Score	Comment	Score	Comment	Score	Comment	Score	
NPBT	red-auburn, little dull, little curly, approach tippy, even	80 ± 0.00 c	floral, fragrant and lasting	88 ± 0.53 b	strong, thick, umami, little bitter	84 ± 0.87 b	orange-red, bright	88 ± 0.53 ab	fat and bold, little dull, even	80 ± 0.00 c	84.2 ± 0.47 c
LPBT	red-auburn, yellow and white, curly, tippy, even	87 ± 0.40 a	flowery and honey aroma, fragrant and lasting	90 ± 0.56 a	umami, brisk, more mellow and thick, sweet after taste	88 ± 0.50 a	deeply orange, bright	87 ± 0.46 b	slightly curly, soft and bright, even	87 ± 0.35 a	88.25 ± 0.44 a
HPBT	red-auburn, yellow and white, more curly, more tippy, more even	84 ± 0.50 b	more flowery and honey aroma	86 ± 0.56 c	mellow and thick, sweet, more umami and brisk	88 ± 0.62 a	amber, bright	89 ± 0.00 b	curly, red and bright, even	85 ± 0.20 b	86.35 ± 0.43 b

The letters (a, b, c) indicated statistical significance (*p* < 0.05) between the same columns.

**Table 2 foods-12-01737-t002:** The qualitative results for the volatile components in the three beauty teas.

No.	Compounds	Odour Description	Concentration (µg/kg Dry Weight of Tea Leaves)		
			NPBT	LPBT	HPBT
1	1,6-octadien-3-ol, 3,7-dimethyl-(linalool)	floral, fruity	22.55 ± 3.88 ^a^	23.26 ± 6.45 ^a^	22.97 ± 6.25 ^a^
2	1,5,7-octatrien-3-ol, 3,7-dimethyl-(dihydrolinalool)	floral, fruity	43.11 ± 8.09 ^b^	61.55 ± 15.36 ^b^	104.06 ± 23.12 ^a^
3	2h-pyran-3-ol, 6-ethenyltetrahydro-2,2,6-trimethyl-	citrus-like, floral, green	2.59 ± 0.62 ^b^	3.05 ± 1.05 ^b^	7.53 ± 0.58 ^a^
4	geraniol	floral, sweet	14.79 ± 1.35 ^a^	11.29 ± 2.93 ^a^	9.31 ± 2.47 ^a^
5	1,6,10-dodecatrien-3-ol, 3,7,11-trimethyl-, [S-(Z)]-	floral	2.02 ± 0.22 ^a^	2.14 ± 0.19 ^a^	NA
6	cedrol	cedarwood- like	1.09 ± 0.11 ^a^	NA	NA
7	cis-linalool oxide	sweet, floral, cream	12.27 ± 3.54 ^b^	15.6 ± 4.92 ^b^	30.48 ± 5.67 ^a^
8	trans-linalool oxide (furanoid)	sweet, floral	3.15 ± 2.38 ^b^	4.56 ± 1.44 ^b^	10.35 ± 0.84 ^a^
9	benzaldehyde	sweet, fruity	23.14 ± 3.17 ^b^	26.33 ± 5.96 ^b^	50.99 ± 7.4 ^a^
10	benzeneacetaldehyde	floral, fruity, sweet	5.03 ± 0.94 ^a^	2.98 ± 1.62 ^a^	6.32 ± 1.89 ^a^
11	decanal	sweet, floral, citrus-like	4.47 ± 0.48 ^a^	5.23 ± 0.43 ^a^	5.46 ± 0.55 ^a^
12	1-cyclohexene-1-carboxaldehyde, 2,6,6-trimethyl-(β-cyclocitral)	fruity, fresh and sweet	2.44 ± 0.22 ^b^	4.4 ± 0.66 ^a^	4.33 ± 1.08 ^a^
13	3-hydroxymandelic acid, ethyl ester, di-TMS	bitter almond	2.68 ± 1.11 ^a^	4.56 ± 2.23 ^a^	9.13 ± 4.69 ^a^
14	ethyl 2-(5-methyl-5-vinyltetrahydrofuran-2-yl)propan-2-yl carbonate	fruity	12.77 ± 2.88 ^b^	16.58 ± 5.16 ^b^	38.19 ± 7.87 ^a^
15	propanoic acid, 2-methyl-, hexyl ester	fruity	2.22 ± 0.26 ^a^	2.04 ± 0.45 ^a^	NA
16	butanoic acid, 3-hexenyl ester, (E)-	sweet, fruity	4.54 ± 0.23 ^a^	3.35 ± 0.63 ^a^	3.47 ± 0.88 ^a^
17	methyl salicylate	holly oil, minty	6.54 ± 0.37 ^a^	5.97 ± 1.66 ^a^	6.08 ± 0.15 ^a^
18	butanoic acid, hexyl ester	sweet, fruity	6.61 ± 0.48 ^a^	5.79 ± 0.8 ^a^	5.55 ± 0.57 ^a^
19	cis-3-hexenyl isovalerate	sweet, apple-like, grassy green	15.13 ± 1.51 ^a^	15.62 ± 2.51 ^a^	5.61 ± 1.52 ^b^
20	butanoic acid, 2-methyl-, hexyl ester	sweet, fruity	20.93 ± 0.83 ^a^	22.23 ± 3.19 ^a^	4.52 ± 1.11 ^b^
21	2,6-octadienoic acid, 3,7-dimethyl-, methyl ester	floral, herbal smell, citrus-like	2.65 ± 0.47 ^a^	2.66 ± 0.49 ^a^	NA
22	hexanoic acid, 3-hexenyl ester, (Z)-	fresh and sweet, floral	12.29 ± 1.96 ^a^	12.59 ± 2.43 ^a^	4.23 ± 0.47 ^b^
23	hexanoic acid, hexyl ester	bean-like, fruity	13.12 ± 3.16 ^a^	13 ± 2.1 ^a^	3.54 ± 0.4 ^b^
24	hexanoic acid, 2-hexenyl ester, (E)-	fruity	3.85 ± 0.66 ^a^	3.44 ± 0.62 ^a^	NA
25	bicyclo [2.2.1]hept-2-ene, 1,7,7-trimethyl-	-	6.85 ± 0.6 ^a^	8.98 ± 2.38 ^a^	5.7 ± 1.11 ^a^
26	.alpha.-cubebene	-	1.26 ± 0.35 ^a^	NA	NA
27	.alfa.-Copaene	wormwood smell	2.43 ± 0.57 ^a^	2.61 ± 0.27 ^a^	NA
28	1h-3a,7-methanoazulene, 2,3,4,7,8,8a-hexahydro-3,6,8,8- tetramethyl-, [3R-(3.alpha.,3a.beta.,7.beta.,8a.alpha.)]-	woody	2.67 ± 0.51 ^b^	3.27 ± 0.04 a ^b^	3.94 ± 0.7 ^a^
29	caryophyllene	spicy, woody, citrus-like	18.1 ± 1.34 ^a^	10.5 ± 0.75 ^b^	NA
30	humulene	spicy, woody, citrus-like	2.01 ± 0.29 ^a^	NA	NA
31	.alpha.-Muurolene	pine-like, citrus-like	1.94 ± 0.52 ^a^	1.92 ± 0.53 ^a^	NA
32	naphthalene, 1,2,3,5,6,8a-hexahydro-4,7-dimethyl-1-(1-methylethyl)-, (1S-cis)-	woody	12.96 ± 2.61 ^a^	NA	2.32 ± 0.39 ^b^
33	.alpha.-calacorene	woody	1.6 ± 0.49 ^a^	NA	NA
34	.tau.-muurolol	pine-like, citrus-like	1.96 ± 0.28 ^a^	1.66 ± 0.22 ^a^	NA
35	.gamma.-muurolene	pine-like, citrus-like	NA	1.5 ± 0.06 ^a^	NA
36	naphthalene, 1,2,4a,5,8,8a-hexahydro-4,7-dimethyl-1-(1-methylethyl)-, [1S-(1.alpha.,4a.beta.,8a.alpha.)]-	woody	NA	14.25 ± 1.43 ^a^	NA
37	hexane, 3,3-dimethyl-	-	8.01 ± 0.85 ^a^	NA	NA
38	cyclopentasiloxane, decamethyl-	-	14.42 ± 4.53 ^a^	NA	NA
39	undecane, 2,6-dimethyl-	-	2.1 ± 0.63 ^a^	NA	NA
40	cyclohexasiloxane, dodecamethyl-	-	6.8 ± 2.23 ^a^	NA	NA
41	dodecane, 1-iodo-	-	NA	NA	1.98 ± 0.26 ^a^
42	1-oxaspiro [4.5]dec-6-ene, 2,6,10,10-tetramethyl-	fruity, sweet, woody	NA	NA	2.26 ± 0.56 ^a^
43	benzene, 1,2,3-trimethyl-	-	15.17 ± 2.39 ^a^	NA	NA
44	benzene, 1,2,3,5-tetramethyl-	-	6.61 ± 1.28 ^a^	4.14 ± 1.86 ^a^	8.38 ± 3.05 ^a^
45	benzene, 1,2,4-trimethyl-	-	NA	NA	22.81 ± 4.82 ^a^
46	trans-.beta.-Ionone	violet smell, fruity, woody	1.83 ± 0.26 ^b^	1.27 ± 0.8 ^b^	3.56 ± 0.13 ^a^
47	(R,S)-5-Ethyl-6-methyl-3E-hepten-2-one	fresh and sweet, green	NA	NA	13.85 ± 2.76 ^a^
48	furan, 2-pentyl-	fruity, green	NA	11.78 ± 2.55 ^a^	NA
49	thiourea, N-methyl-N’-phenyl-	-	5.75 ± 0.99 ^a^	6.26 ± 1 ^a^	NA

“-” represents that no odor description, and threshold information was found in the literature. “NA” represents that the compound was not detected in sample. The letters (a, b) in the same row indicated statistical significance (*p* < 0.05). The odor threshold (OT) of volatile compounds (Appendix A) in water referred in the relevant literature [11,12,13].

## Data Availability

The data presented in this study are available on request from the corresponding author.

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
