# Peer review of "Chemical and Quality Analysis of Beauty Tea Processed from Fresh Leaves of Tieguanyin Variety with Different Puncturing Degrees"

_foods, 2023, doi:10.3390/foods12091737_

Round 1
Reviewer 1 Report
The authors study metabolites generated during beaty tea production and wether the state of the fresh tea leave have any influence on them.
Comments:
-Do not use etcs
-Define all acronyms before using them such as OAV in the abstract
-It is unclear where you are applying each statistical test unless you go thorugh all the manuscript. In the statistics section, explain where and why you are using each of the tests. For instance, why are you using PLS regression? It appears that you just use it to find correlations. Is the Y numeric or categorical in this case? It is also unclear when you use ANOVA. For instance, if you compare the content of a metabolite between two teas, you need to use wilcox or t test
-Across the manuscript, you speak of significant differences but you do not provide any p value...
-Avoid the use of "obvious differences" and stuff like that because that is completely subjective, and unless you provide a p value, differences are not significant and therefore not important scientifically speaking
-You are not proving that the models generated by PLS-DA or PLSR are actually any good because first, you need to tell how are you training the models: are you using all samples? If you want to build a predictive model, you need to use 70-80% of random samples to train the model, test it on the remaining random samples and built a ROC that will tell whether your model is actually any good, incluiding predictive capability. As you are presenting your data, you can only tell which metabolites are related to which tea
-The pathway section needs more explaining. What are you actually talking about? Plant pathway or chemical reaction during processing? One has little to do with the other and I doubt that you can find the latter information in Kegg
-I can get behind using PLS-DA to find discriminant metabolites but why are you using PLSR just to find correlations? Why not Pearson or Spearman correlation? If you are going to use a machine learning method explain why this one and not for instance a multinomial regression (if Y categorical) or simply a linear regression (if Y numeric)
-Why you are only showing VIP values for volatile compounds? You must have these values for non-volitile ones
Reviewer 2 Report
The manuscript applied metabolomics analysis to evaluate the characteristic metabolites of beauty tea fresh leaves and their dry tea with different degrees of leaf hopper puncturing. Some revisions are necessary .
1. Please rewrite abstract in a more concise way.
2. Improve Introduction section and clarify better the aim of the work.
3. Figure 4 is not clear too much image. Please revise.
4. Figure 6 is not clear too much image. Please revise.
5. Figure 7 and 8 are not clear too much images. Please revise.
6. Table 2 is not easy to read please try to divide it.
7. The content of conclusion section are quite similar to discussion. Please write a conclusion in which highlight the obtained results.
Round 2
Reviewer 1 Report
The authors have answered my concerns